# Mitochondrial and Endoplasmic Reticulum Alterations in a Case of Amyotrophic Lateral Sclerosis Caused by TDP-43 A382T Mutation

**DOI:** 10.3390/ijms231911881

**Published:** 2022-10-06

**Authors:** Giada Zanini, Valentina Selleri, Milena Nasi, Anna De Gaetano, Ilaria Martinelli, Giulia Gianferrari, Francesco Demetrio Lofaro, Federica Boraldi, Jessica Mandrioli, Marcello Pinti

**Affiliations:** 1Department of Life Sciences, University of Modena and Reggio Emilia, Via G. Campi 287, 41125 Modena, Italy; 2National Institute for Cardiovascular Research, Via Irnerio 28, 40126 Bologna, Italy; 3Department of Medical and Surgical Sciences for Children and Adults, University of Modena and Reggio Emilia, 41125 Modena, Italy; 4Department of Biomedical, Metabolic and Neurosciences, University of Modena and Reggio Emilia, Via G. Campi 287, 41125 Modena, Italy

**Keywords:** TDP-43, amyotrophic lateral sclerosis, mitochondria, mtDNA

## Abstract

Amyotrophic lateral sclerosis is the most common form of motor neuron disease. Mutations in *TARDBP*, the gene encoding the RNA-binding protein TDP-43, are responsible for about 5% of familial ALS. Here we report the clinical and biological features of an ALS patients with pA382T mutation in TPD-43 protein. Disease began with right hand muscles weakness, and equally involved upper and lower motor neuron with a classic phenotype, without cognitive impairment. While a family history of neurological diseases was reported, there was no evidence of familial frontotemporal dementia. Cultured fibroblasts from the patient were characterized by profound alterations of cell proteome, which impacts particularly the mitochondrial metabolic pathways and the endoplasmic reticulum. TDP-43 levels were similar to control, healthy fibroblasts, but a higher fraction localized in mitochondria. Mitochondrial network appeared fragmented, and the organelles smaller and more spheric. In agreement with impaired proteome and morphology of mitochondria, basal cell respiration was reduced. Mitochondrial DNA levels appeared normal. However, a higher amount of mitochondrial DNA was present in the cytosol, suggesting a pronounced mitochondrial DNA misplacement which can promote a pro-inflammatory response mediating by cGAS/STING. Thus, this case report further expands the clinical and pathological phenotype of A382T mutation.

## 1. Introduction

TAR DNA Binding Protein (TARDBP or TDP-43) is an RNA-binding protein encoded by *TARDBP* gene located on chromosome 1 whose mutations and pathological neuronal cytoplasmic ubiquitinated inclusions are associated with amyotrophic lateral sclerosis (ALS) with or without frontotemporal dementia (FTD). Mutations in the *TARDBP* gene were first described as a cause of ALS in 2008 [1]. Since then, approximately 50 *TARDBP* mutations have been described, and this gene is now known to be responsible for up to 5% of familial ALS cases and less than 1% of sporadic cases [2]. The majority of the mutations affect the C-terminal glycin-rich domain of the protein [3,4]. Phenotypically, *TARDBP* mutations are associated with a wide range of clinical manifestations [5], including cases with slow, medium, and fast disease progression [6], bulbar or upper limb onset, with or without cognitive and/or behavioral impairment [7]. The frequency and kind of *TARDBP* mutations are not homogeneous among different populations with higher frequency in Southern Europe (up to three times higher, especially in France and in Italy) than in other Caucasian populations [8] where an additional founder effect for A382T mutation has been described in Sardinia [9]. More specifically, the A382T mutation has been associated with a higher penetrance in men than in women, predominant upper limb onset, mean age at onset between 60 and 65 years and presence of FTD in ~10% of cases [8]. Less frequently, the same mutation has been detected in patients with parkinsonisms, with or without overlapping of pyramidal and cognitive clinical signs [10].

The effects of TDP-43 pathogenetic mutations on TDP-43 functions are quite heterogenous. A large fraction of the pathogenetic TDP-43 mutations, including A382T, has been shown to decrease the nuclear localization of the protein, to increase its phosphorylation status, and to favour the formation of C-terminal fragments deriving from proteolytic digestion of the full-length proteins [11]. These fragments can form insoluble aggregates; however, the A382T mutation is not associated with an increased insolubility of the full-length protein, nor reduces the solubility of the C-terminal fragments. While some pathogenic mutations, such as M337V, spontaneously mislocalizes to the cytoplasm and form insoluble aggregates, A382T mutation leads to aggregate formation only in the presence of ER stress. A minority of mutations, which does include A382T, also increases the protein half-life [11].

Here, we describe an ALS family due to an A382T mutation in the *TARDBP* gene and the study of mitochondrial dysfunctions in patient’s derived fibroblasts. Additionally, we review the available literature to discuss the possible contribution of the A382T mutation to the pathogenesis of ALS.

## 2. Results

### 2.1. Clinical Findings

A 43-year-old healthy man (Figure 1, IV-1, the proband) presented with an eleven-month history of right-hand muscle weakness and atrophy, followed by the appearance of widespread cramps and fasciculations and mild walking impairment; Table 1 summarizes his main clinical signs and symptoms. His medical and social histories were uninformative. The neurological examination showed moderate weakness of the right upper limb, prominent atrophy in intrinsic hand muscles, mild and distal weakness in lower limbs, widespread fasciculations and brisk tendon reflexes in both upper and lower limbs. Babinski and Hoffmann’s signs were absent. Brain and cervical Magnetic Resonance imaging were normal. Electrophysiologic studies revealed active denervation and reinnervation potentials in three body regions. Motor conduction time as measured by transcranial motor evoked potentials was normal; cortical silent period was shorter when recorded at the right upper limb. Cerebrospinal fluid (CSF) analysis was unremarkable. Neuropsychological testing did not show evidence of cognitive impairment. Respiratory function was normal, and his revised ALS functional rating scale (ALSFRS-R) score was 41/48. The diagnosis of ALS was established based on the EL Escorial Revised Criteria. Three years after symptom onset the patient presented dysarthria and dysphagia, pseudobulbar affect, severe upper and lower limbs weakness and atrophy, more marked at the distal level, widespread fasciculations, brisk tendon reflexes, with initial respiratory involvement that required non-invasive ventilation a few months before (ALSFRS-R score:16/48). He underwent tracheostomy and percutaneous endoscopic gastrostomy 4 months later.

The patient reported a family history of ALS. The proband’s mother (Figure 1, II-2) developed ALS when she was 64 years old, 6 years before her son. A maternal aunt (Figure 1, II-4) is affected by multiple sclerosis. The maternal grandmother (Figure 1, III-2) died at 65 years of age due to a brain tumor. The maternal grand-grandfather was affected by epilepsy (Figure 1, IV-2).

Informed consent from the patient was obtained for genetic analysis. Repeat-primed PCR did not detect a pathogenic C9ORF72 expansion. Next-generation sequencing (NGS) revealed a heterozygous 1144G>A substitution in the *TARDBP* gene, causing A382T amino acid substitution (For details, see supplementary methods). The proband’s mother (Figure 1, II-2) finally resulted to carry the same 1144G>A mutation. She died at the age of 72 years.

### 2.2. Biological Assessments

#### 2.2.1. Alterations of the Proteome of Fibroblasts from the Proband

We set up a primary culture of fibroblasts from a skin biopsy of the proband and compared their features with fibroblasts from a healthy donor matched for sex and age. Patient’s fibroblasts grew normally, with a proliferation rate similar to healthy donor’s cells.

To determine the effects of A382T mutation on cell functions, we first compared the proteome of pathological and wild type fibroblasts. We identified a total of 2056 proteins (Appendix A), whose 924 were identified with at least two peptides, and considered as reliable identifications [12]. One hundred eighty-two proteins were differentially expressed proteins (DEPs) between ALS and CTRL fibroblasts; 116 resulted up-regulated and 66 down-regulated (Figure 2A,C). The Figure 2B shows the number of DEPs in different extra- and sub-cellular compartments: more than one-third of differentially expressed proteins (DEPs) are located in the mitochondrion (i.e., 66/182, 36%) suggesting a profound impact of A382T r mutation on the organelle, followed by cytosolic polypeptides (i.e., 40/182; 22%). To discover which pathways were preferentially regulated in ALS, we separately analyzed the up-regulated (i.e., 116/182) and down-regulated DEPs (i.e., 66/182) using Metascape. Figure 2D,E show the top 20 clusters obtained from up- and down-regulated DEPs, respectively. Proteins with high expression levels in ALS were highly enriched in genes related to the generation of precursor metabolites and energy, carboxylic acid catabolic process, valine, leucine and isoleucine degradation and electron transport chain (Figure 2C). In contrast, down-regulated DEPs in ALS were significantly enriched in proteins related to actin filament-based process, ITGA5-ITGB1-FN1-TGM2 complex, cell-cell adhesion and cell-matrix interaction (Figure 2D). Manual inspection of DEPs also revealed the presence of the RNA Helicase DDX3X, a repressor of C9ORF72 Repeat-Associated Non-AUG aberrant translation [13], among upregulated proteins. Taken together, these data suggest that ALS fibroblasts may have altered mitochondrial function and dysregulation of the cytoskeleton, two common features reported in different neurodegenerative diseases, including ALS [14].

We selected a subset of DEPs to confirm results by immunoblot. As shown in Figure 2F, Malate Dehydrogenase 2 (MDH2), and mitochondrial Lon protease (LONP1) were significantly overexpressed in A382T fibroblasts. TDP-43 levels were not significantly different from control, but its distribution was altered, as cells from ALS patients showed a higher relative level of mitochondrial TDP-43 and a lower cytosolic fraction than control cells (Figure 2G).

#### 2.2.2. Morphological, Ultrastructural and Functional Alteration of Fibroblasts

Since the altered expression of cytoskeleton proteins can induce a change in cell shape, we have analysed the morphology of CTR and ALS fibroblasts by light microscopy (Figure 3A). Control cells showed a spindle-shaped morphology, whereas pathological fibroblasts were more flattened on the surface with abundant cytoplasm. We quantified the area and perimeter of cells, which were significantly higher in ALS than CTRL fibroblasts. Furthermore, aspect ratio and circularity were significant changes in pathological cells, whilst no difference was found for solidity (Figure 3B).

Figure 3C,D show the main ultrastructural abnormalities observed in ALS fibroblasts. Mitochondria showed an elevated heterogeneity in morphology: some were particularly enlarged with distorted or excessively branched cristae, and others were characterized by reduction of many cristae (Figure 3C, insert). Moreover, in comparison with CTRL fibroblasts (Figure 3D upper panel), the endoplasmic reticulum cisterns in ALS cells were swollen with an irregular shape, suggesting an accumulation of unfolded/misfolded polypeptides in the lumen (Figure 3D, lower panel).

Confocal microscopy confirmed ultrastructural observations. Mitochondria of fibroblasts from the proband are more fragmented and reveal a different shape of the mitochondrial network (Figure 3E). While the total volume of mitochondria was like healthy fibroblasts, mitochondria from ALS patient had a lower mean volume, were more spheric, and with a lower number of branches/mitochondria, suggesting the presence of a more fragmented mitochondrial network (Figure 3F).

In agreement with the profound morphological alterations and aberrant protein expression, basal respiration was lower than wild type cells, although maximal respiration, ATP-linked and uncoupled respiration did not show a significant difference (Figure 3G,H). Mitochondrial superoxide levels, as measured by mitoSOX Red fluorescence, were 50% higher in ALS fibroblasts (Appendix A), suggesting the presence of oxidative stress. However, we did not detect any difference in the expression of representative genes related to stress response in mutant and wild type fibroblasts (Appendix A).

It has been previously observed that TDP-43 mutation can determine mitochondrial DNA (mtDNA) release from the organelle to the cytosol. To test if this also occurs in the proband’s fibroblasts, we first revealed the presence of extramitochondrial dsDNA in the cytoplasm of fibroblasts, and we found that dsDNA was present both in wild type and mutant fibroblasts (Figure 3I). Then, we measured the absolute level of mtDNA in the cell, and its intracellular distribution. MtDNA levels were similar in ALS and CTRL fibroblasts (Figure 3L, left), but we did find a higher level of mtDNA in the cytosol of ALS fibroblasts, suggesting that a pronounced cytosolic misplacement of mtDNA was present in ALS mutant cells (Figure 3L, right). Interestingly, we also observed a higher level of stimulator of interferon gene (STING), which senses cytosolic DNA and activates an inflammatory signaling program, in fibroblasts from ALS patient.

As mtDNA misplacement and the activation of the cGAS/STING pathway can cause lead to the release of pro-inflammatory cytokines, we determined the levels of tumor necrosis factor-alpha (TNF-α), interleukin-1beta (IL-1β), interleukin-6 (IL-6), interleukin-8 (IL-8) in the cell supernatant. No significant production of TNF-α and IL-1β has been observed in both control and ALS fibroblasts. Conversely, IL-6 and IL-8 were lower in the supernatant of ALS cells (Appendix A).

## 3. Discussion

TDP-43 mislocalization and dysfunction have been identified as key pathological features in ALS and FTD [15]. Although the protein exerts several crucial functions for cell survival and functionality, including regulation of transcription and metabolism, its role in the pathogenesis of ALS has not been fully elucidated and whether a loss or a gain of function plays a role in neurodegeneration remains controversial [16].

In addition to RNA metabolism impairment [17], in ALS TDP-43 has been linked to R-loop modulation [18], cell response to stress through stress granules [19], Bcl-2 mediated ER Ca^2+^ signaling dysregulation [20], and impairment of mitochondrial activity [21]. 

Morphological and functional alterations have been observed in muscle and brain from sporadic patients and in fibroblasts and monocytes from ALS patients carrying mutations in several genes associated with ALS (e.g., *SOD1*, *TARDBP*, and *VCP*) [22].

It has been previously observed that A382T can be associated with FTD in 10% of cases, with typical Parkinson’s disease, as well as with atypical (PSP-like or CBD-like) parkinsonisms [23]. However, the family of the proband did not record PD or FTD cases, but registered a case of MS, one of epilepsy, and one case of brain tumour, possibly expanding the number of neurodegenerative diseases that could be associated with A382T substitution. Unfortunately, segregation study was not possible due to the unavailability of blood samples of affected family members. The correlation between MS and ALS is complex and controversial, particularly as far as familial cases are concerned. Genome-wide association meta-analysis of SNPs did not evidence any overlap in genetic susceptibility between MS and ALS [24]. Another study failed to provide evidence of a correlation between A382T mutation and MS susceptibility in Sardinians, where A382T mutation is observed in 30% of ALS patients, suggesting that this mutation does not play a direct role in MS susceptibility. However, large epidemiological studies suggested an association between MS and ALS [25,26], with an increased risk of MS among children of ALS patients [27,28]. If co-occurrence of ALS and MS is often due to diagnostic uncertainty [29], in validated cases of co-occurrence, the MS-related neuroinflammatory symptoms occurred years before ALS onset, leading to hypothesize that, in a small subgroup of ALS patients with defined clinical characteristics, neurodegeneration may be triggered by preceding neuroinflammation around the motor unit [29]. Thus, it could be possible that the inflammatory pathways activated by mitochondrial dysfunctions can contribute to neuroinflammation observed in MS, and in ALS. 

In this regard, our study unrevealed several alterations in the mitochondria of proband’s fibroblasts, which resulted different in shape and ultrastructure from wild type cells. While some alterations have been previously observed [21], other morphological changes are novel. 

The alteration in the shape of A382T fibroblasts, never described before, likely reflects the changes in the expression of cytoskeleton-related pathways, and the impairment in the ER and mitochondrial ultrastructure we observed. Although only a few proteins in the ER appeared differentially expressed in ALS fibroblasts, the ER appeared enlarged, irregularly shaped and swollen, likely because of the accumulation of misfolded proteins, and is suggestive of ER stress. ER stress has been often observed in ALS, and the presence of abnormal protein aggregates and/or of inclusions of misfolded proteins is a common feature of ALS; the presence of an enlarged ER could be of particular interest because, differently from other pathogenetic mutations like M337V, A382T causes the formation of cytoplasmic TDP-43 aggregates only in the presence of ER stress [20]. As the presence of enlarged ER was not reported before in A382T fibroblasts, these observations further underlined the heterogeneity in the phenotypical manifestations of A382T r mutation. 

We further expanded previous observations concerning mitochondrial alterations in A382T mutants previously reported [21]. Our data show that the accumulation of mutant TDP-43 in the mitochondria deeply altered proteome, morphology, and functionality of the mitochondria in fibroblasts.

We first confirmed that A382T mutation determines a higher level of TDP-43 accumulation in the mitochondria. It has been previously shown that mitochondrial import of TDP43 depends on the presence of internal mitochondrial localization motifs [30,31]. Wang et al. identified six putative internal mitochondrial import motifs (M1-M6) in TDP-43 and demonstrated that at least two of them (M1 and M3) are functional, being M1 the most efficient. Neither M1 nor M3 include the codon 382 [31]. However, the A382T mutation falls in the M6 motif. Although there is no direct proof that M6 is functional, we can speculate that A382T mutation in M6 makes it more efficient. An alternative, non-mutually exclusive hypothesis could be that this mutation indirectly modifies the accessibility of M1 or M3 sequence during TDP-43 import.

Proteomic analysis revealed a profound alteration in the relative levels of proteins related to the tricarboxylic acid (TCA) cycle and oxidative phosphorylation system (OXPHOS), suggesting that an enhanced localization of a mutated form of TDP-43 in the mitochondria leads to an imbalance of the biological processes crucial for energy production, as already demonstrated for other ALS genes [32]. This agrees with the reduction of basal respiration of A382T fibroblasts, and partially discordant with the observations made by Onesto and colleagues [21], who did not find significant differences in OCR between mutant and wild type fibroblasts. This discrepancy suggests that the impact on mitochondria of this mutation could greatly vary, and likely could contribute to the variable penetrance of the mutation. Concerning mitochondrial network, we performed the analysis in 3D, and the presence of a more fragmented network with spheric mitochondria a lower mean volume of the organelle agrees with previous observations reporting a more rounded shape of mitochondria, associated with a fragmented network observed in 2D confocal images. In line with these profound derangements of mitochondrial shape and function, we also observed an increase in the levels of mitochondrial ROS which can contribute to cell damage and ALS onset, as observed for other ALS-causing genes, an in particular SOD1 [33,34].

The quantification of the total amount of mtDNA showed a similar level in mutant cells compared to wt cells, even if the analysis of its intracellular distribution showed a double level of mtDNA in the cytosol of ALS cells, suggesting that a cytosolic leakage occurred. It has been previously shown that mutations of TDP-43 cause a release of mtDNA leakage to the cytosol, with activation of the cGAS/STING pathway [35]. This pathway has been shown to trigger the inflammatory status observed in ALS patients. Although we could not prove directly that such a pathway is activated also in A382T fibroblasts, the higher level of STING observed by proteomic analysis suggests this higher level of mtDNA in the cytosol can activate a pro-inflammatory response. However, we could not observe a significant release of pro-inflammatory cytokines in resting conditions, likely because no pro-inflammatory stimulus has been used in our experimental setting.

Among upregulated proteins, LONP1 and DDX3X are of particular interest. We and others have previously shown that LONP1 is crucial for the proper folding of mitochondrial proteins, for degradation of misfolded and damaged proteins, and for maintenance of mtDNA [36,37,38]. The upregulation of LONP1 has been described in the presence of proteotoxic or oxidative stress. Thus, the higher level of LONP1 found in ALS could be part of a response to stress caused by mt protein imbalance. As LONP1 is also capable of binding mtDNA [37,38], its higher levels can also be a response mechanism for sequestering mtDNA in the mitochondria and limiting mtDNA leakage to the cytosol.

Finally, the observation that the mitochondrial RNA Helicase DDX3X—a repressor of C9ORF72 aberrant translation [13]—is downregulated in these cells suggests another important overlap between two different genes causative of ALS, namely TARDBP and C9ORF72. It has been shown that when DDX3X is reduced, aberrant accumulation of dipeptide repeat (DPR) proteins in patients with GGGGCC repeat expansion in C9ORF72 occurs and leads to cell toxicity. It is theoretically possible that DDX3X downregulation can contribute to aberrant protein translation in A382T cells. However, the mechanism(s) that connects TDP-43 mutation with DDX3X down-regulation is not clear and need further investigation.

In view of the key role of TDP-43 in the pathomechanisms of the vast majority of ALS, the discovery of new factors playing a part in its regulatory activity may provide potential therapeutic targets for ALS, even independently from TARDBP mutations.

## 4. Materials and Methods

### 4.1. Ethics Statement

Human skin fibroblasts were generated from skin biopsies of one patient affected by ALS and carrying the A382T mutation in the gene coding for TARDBP, and two healthy donors matched for sex and age. All participants provided written informed consent for the collection of skin biopsies and the use of skin fibroblast lines for research purposes. The present study was approved by the Ethical Committee of Area Vasta Emilia Nord (number 366/2018, 19 September 2018).

### 4.2. Isolation and Culture of Fibroblasts

Fibroblasts were isolated from biopsies taken from the arm of the ALS patient and of a healthy male aged 46, after informed consent in accordance with the World Medical Association’s Declaration of Helsinki. Cells were maintained in Dulbecco’s modified Eagle’s medium with the addition of 10% fetal bovine serum, 100 μg/mL streptomycin, and 100 U/mL penicillin. Cells were cultured in a humidified incubator with 5% CO_2_ at 37 °C and confirmed to be free of mycoplasma contamination. All experiments were performed between passages 3–6.

### 4.3. Microscopy and Analysis

#### 4.3.1. Light Microscopy

Cells from ALS and CTRL sample were seeded in plates and cultured for 72 h as described above, then fixed in 4% *v*/*v* paraformaldehyde in DPBS for 10 min, washed with DPBS and stained with 1% toluidine blue and washed with DPBS. Fibroblasts were photographed using a Nikon DS-Fi1 (Nikon Corporation, Tokyo, Japan) camera coupled with a Zeiss Axiophot light microscope.

Cell shape has been characterized through different shape descriptors, i.e., area (total area of each cell), perimeter (cell external perimeter), aspect ratio (the ratio between the major and minor axis of a cell), circularity [4π (area)/(perimeter^2^)], and solidity (density of the cell). Image analysis was performed on 120 cells, repeated three times for each sample, by using ImageJ v.1.52p software. Results are shown as mean values ± standard errors (SE). Statistical analyses were performed using GraphPad Prism 8.0 software. Differences were considered significant for *p* < 0.05.

#### 4.3.2. Transmission Electron Microscopy

Cells at confluence were detached with scraper and embedded as already described [39]. Ultrathin sections (60 nm) were cut and mounted on 150 mesh copper grids (Electron Microscopy Sciences, Hatfield, PA, USA). After air drying, the grids were observed by FEI NOVA NanoSEM 450.

### 4.4. Proteomic Analysis

#### 4.4.1. Sample Preparation

Frozen cells were lysed in modified RIPA buffer (1% NP-40; 150 mm NaCl; 1% SDS and protease inhibitor cocktail) and homogenized by a G19 needle and then incubated for 30 min in ice. Cell debris were removed by centrifugation at 14,000 rpm for 10 min at 4 °C. The supernatant was transferred to a clean tube and the protein quantity was estimated by Bradford assay.

One hundred µg of proteins for each replicate, i.e., three biological and three technical replicates for each cell line, were reduced and alkylated by 5 mM dithioerythritol and 15 mM iodoacetamide, respectively. Trypsin (Promega, Madison, WI, USA) digestion was performed on solution buffered in 50 mM NH_4_HCO_3_ overnight at 37 °C using an enzyme-to-protein ratio of 1:50 (*w*/*w*). All the reagents were purchased from Sigma-Aldrich (Merk KGaA, Darmstadt, Germany) unless otherwise stated.

Mass spectrometric analysis of samples was performed with a Q Exactive (Thermo Fisher Scientific, Waltham, MA, USA), as previously described [40].

#### 4.4.2. Protein Identification and Quantification

MS/MS ions search was performed using Comet search engine (v. 2021.01 rev. 0) integrated in Tran-Proteomic Pipeline (v. 6.0.0) converting raw MS/MS using default settings of msConvert ProteoWizard (v.3.0.1908) to MZML file. Human reference protein datasets were downloaded from Uniprot (UP000005640), integrated with common protein contaminant cRAP (v. 2012.01.01). Sequences were reversed to generate decoy database. The selected parameters for protein identification were: (i) at least 1 unique peptide; (ii) static modification carbamidomethyl on cysteines (+57.021 Da), dynamic modifications oxidation on methionine (+15.995 Da) and deamidation on asparagine and glutamine (+0.984 Da); (iii) precursor mass tolerance of 10 ppm, fragment mass tolerance of 0.02 Da; (iv) the maximum of missed trypsin cleavage sites of 1; (v) the minimum peptide length of 7; (vi) peptide were performed by PeptideProphet the output refined using iProphet excluding number of sibling peptides (NSP) model; (vii) protein validation were performed using ProteinProphet on iProphet output. Briefly, iProphet peptide results (.pepxml) were imported in Skyline-daily (v.21.0.9.139) as already described [40], to generate spectral libraries. Parameters settled were: (i) 0.99 as spectra cut-off score; (ii) precursor ion charge 2+, 3+, 4+; (iii) MS1 filters were set to “use high selectivity extraction” with a resolving power of 60,000 at 300 m/z; (iv) repeated and duplicate peptides were removed; (v) Fasta files containing proteins with 1% FDR were imported to Skyline to maintain and fix FDR; (vi) only proteins with at least two peptides were considered for quantitative analysis to avoid incorrect quantification across LC-MS runs [12]. Label free quantification were performed with MSstats (v.4.1.2.1) tool, integrated in Skyline, using Equalize medians normalization method and selecting high quality features. Proteins with log_2_ fold change ± 0.58 and an adjusted *p*-values < 0.05 were considered as differentially expressed proteins (DEPs).

### 4.5. Sub Cellular and Extracellular Localization and Pathway Enrichment Analysis of DEPs

The extracellular and/or subcellular localization of DEPs were performed by interrogating different databases: MatrisomeDB, a database that includes all structural ECM components and proteins that may directly or indirectly interact with the ECM (http://www.pepchem.org/matrisomedb, accessed on 6 April 2022), Human MitoCarta 3.0, a reference inventory of mitochondrial proteins [41], The Human Protein Atlas (https://www.proteinatlas.org) and UniProtKB (http://www.uniprot.org/, accessed on 6 April 2022).

Moreover, we used Metascape (https://Metascape.org/, accessed on 6 April 2022), a web-based portal designed, to find significantly enriched pathway analysis in our target gene list [42]. 

### 4.6. Immunoblot

Immunoblot was performed as previously described [43]. Total cell lysates were prepared in RIPA lysis buffer plus protease inhibitors cocktail (Sigma Aldrich, St. Louis, MO, USA) and phosphatase inhibitors (Sigma Aldrich). Nuclear, mitochondrial, and cytosolic fractions were isolated by using the Cell fractionation kit (Abcam, Cambridge, UK), following provided instructions. Samples were resolved by SDS-PAGE on precast gels (8%, 12%, 4–12%, ThermoFisher Scientific, Waltham, MA, USA) and transferred to nitrocellulose membranes (Bio-Rad Laboratories, Hercules, CA, USA), which were then immunoblotted. The following primary antibodies were used: anti-TDP-43 (1:1000; Wuhan Fine Biotech, Wuhan, China, FNab08577), anti-tubulin (1:500; Santa Cruz Biotechnology, Dallas, TX, USA, sc-5286), anti- LaminB1 (1:1000; Abcam, Cambridge, UK, #ab16048), anti-Tom20 (1:500; Santa Cruz Biotechnology, Dallas, TX, USA, sc-17764), anti-LonP1 (1:1000; Primm, Milan, Italy) [43,44], anti-β-actin (1:1000; Abcam, Cambridge, UK, #ab8227), anti-MDH2 (1:1000; Thermo Fisher Scientific, PA5-21700). The following secondary antibodies were used: HRP-conjugated goat anti-rabbit and HRP-conjugated goat anti-mouse (Bio-Rad Laboratories, #1706515, #1706516). Enhanced Clarity chemiluminescent substrate (Bio-Rad laboratories) was used to detect proteins by using a Chemidoc MP (Bio-Rad Laboratories). Image analysis was performed by Image Lab software v5.2.1.

### 4.7. Confocal Microscopy

Fibroblasts were seeded at 150.000 cells/well in 24-well plates on glass coverslips pre-treated with polylysine (Sigma-Aldrich), fixed with 3.7% formaldehyde (Sigma Aldrich) in PBS, for 9 min, and then with acetone (Sigma Aldrich) for 5 min at −20 °C. After fixation, cells were permeabilized with 0.1% Triton X-100 in PBS for 6 min and blocked with 3% bovine serum albumin (BSA) in PBS for 30 min, at room temperature. Incubation with primary antibodies was performed for 1 h at room temperature. After washing with 3% BSA in PBS, samples were incubated for 1 h at room temperature with secondary antibodies. Samples were then washed in PBS and stained with 0.5 μg/mL 4,6-diamidino-2-phenylindole (DAPI) (Sigma-Aldrich) in PBS for 5 min, washed again in PBS and mounted. Samples prepared for STED microscopy were not incubated with DAPI. Cells were observed and images were acquired with a Leica TCS SP8 confocal laser scanning microscope (Leica Microsystems, Wetzlar, Germany). The following antibodies were used: anti-LonP1 (1:100; Primm), anti-dsDNA (1:100; Santa Cruz Technologies), anti-hMit (1:100; Sigma Aldrich), goat anti-mouse Alexa Fluor 488 (1:200; Thermo Fisher Scientific), goat anti- rabbit IgG (H + L) Alexa Fluor 647 (1:200; Thermo Fisher Scientific). Image analysis was performed with Fiji (ImageJ) and ScanR.

### 4.8. Mitochondrial Network Analysis

Mitochondrial network analysis has been performed on 3D images of ALS and CTRL fibroblasts, with 16 z-stacks per image, using “Mitochondria analysis” v 2.3 plugin of Fiji (ImageJ 1.53q) [45]. Briefly, mitochondria have been identified using the weighted mean thresholding method, after optimization of 3D threshold parameters. Then, morphological (total volume, mean volume per mitochondrion, surface area, and sphericity) and network (number of branches, total branch length, mean branch length), parameters have been automatically calculated. A total of 30 CTRL and 30 ALS cells have been analyzed. Volume is expressed as ml^3^, sphericity is an absolute number ranging from 0 to 1 (with 1= sphere), and mean branch length is expressed in mm.

### 4.9. Oxygen Consumption Rate (OCR) and Extracellular Acidification Rate (ECAR)

XFe96 Analyzer (Agilent Technologies, Santa Clara, CA, USA) was used to assess OCR and ECAR. Cells were plated in a number of 2 × 10^4^ cells/well the day before the assessment, and experiments were performed on a confluent monolayer. Measurements were performed under basal conditions and then after injections of oligomycin (1.0 μM), carbonyl cyanide 4-trifluoromethoxy-phenylhydrazone (FCCP, 1.0 μM), rotenone and antimycin A (0.5 μM of both). Data have been normalized to sample protein concentrations, following instructions provided by the manufacturer.

### 4.10. Quantification of Mitochondrial DNA

Total DNA was extracted from fibroblasts or from mitochondrial and cytoplasmic fractions of fibroblasts using a QIAmp DNA Minikit (Qiagen, Alameda, CA, USA), following manufacturer’s instructions. A droplet digital (dd)PCR assay was used to quantify mtDNA. One microliter of DNA was added to a 20 μL final volume mixture containing 10 μL of 2× ddPCR Supermix for Probes, 1 μL of ddPCR assay for ND4 (UniqueAssayID: dHsaCPE5043566), 1 μL of ddPCR assay for Actb (UniqueAssayID: dHsaCNS141996500) and 7 μL of nuclease-free water (all reagents from Bio-Rad, Hercules, CA, USA). Droplet generation and reading were performed on a Bio-Rad QX200 ddPCR droplet system [46]. Data are expressed as number of copies of mtDNA/nuclear equivalents. When cell subfractions have been analyzed, data have been normalized to values measured in control cells and data from ALS samples are expressed as percentage of control.

### 4.11. Relative Quantification of mRNA Expression

Total RNA was extracted from cells by QuickRNA miniPrep kit (Zymo Research, Irvine, CA, USA) following manufacturer’s instruction, and the amount of RNA was quantified using the NanoDrop ND-1000 (Thermo Fisher Scientific). Then, 1 µg of RNA was reverse transcribed using iScript cDNA synthesis kit (Bio-Rad, Hercules, CA, USA). The CFX96 Touch Detection System (Bio-Rad) was used to quantify mRNA with SYBR Green chemistry. Eleven genes were detected using pre-validated Prime PCR Assay (Bio-Rad): RPS18 was the reference gene, HSP90AA1, HSPA1L, HSPA4, HSPA6, HSPA1A, TLR4, TLR9, IL6, IL18, TGFB1, TGFBR3. Relative expression of mRNA was calculated through ΔΔ-cycle method.

### 4.12. Reactive Oxygen Species Measurement

Mitochondrial ROS (mtROS) levels were quantified by measuring the fluorescence MitoSOX Red (Thermo Fisher Scientific, Inc.) according to the manufacturer’s procedures. 50,000 cells were seeded in a Lumux multiwell-96 (Sarstedt, Nümbrecht, Germany) and incubated with 5 µM MitoSOX Red for 10 min at 37 °C in the dark. Then, cells were washed three times with 1X PBS and cellular fluorescence was measured after 30 min using Fluoroskan FL Microplate Fluorometer and Luminometer (Thermo Fisher Scientific, Inc.) with excitation λ 544 nm, emission λ 590 nm, and time exposure of one second.

### 4.13. Quantification of Cytokines in Cell Supernatants

Four pro-inflammatory cytokines (TNF-α, IL-1β, IL-6 and CXCL-8) have been quantified by using an ELLA Multianalyte assay (Biotechne, San Jose, CA, USA), following provided instructions. Data are expressed as pg/mL.

### 4.14. Statistical Analysis

Statistical analyses were performed using Prism 9.2.1 (GraphPad Software, La Jolla, CA, USA). The results are expressed as the mean ± SEM. Non-parametric Mann-Whitney test was used to compare quantitative variables. *p* value < 0.05 was considered significant.

## Figures and Tables

**Figure 1 ijms-23-11881-f001:**
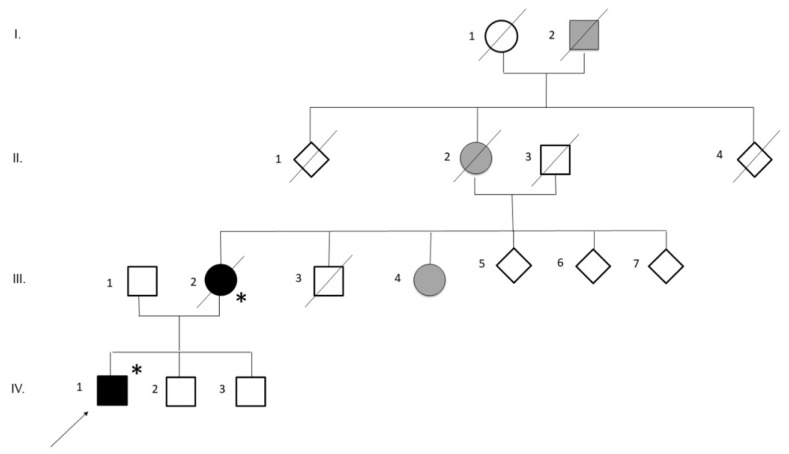
Family pedigree of the family with ALS patients carrying p.Ala382Thr mutation in TARDBP gene. The available DNA samples are indicated by asterisks (*); the proband (IV-2) is marked with an arrow. The filled symbols indicate the affected individuals, with black indicating ALS, grey indicating individuals with other neurological diseases.

**Figure 2 ijms-23-11881-f002:**
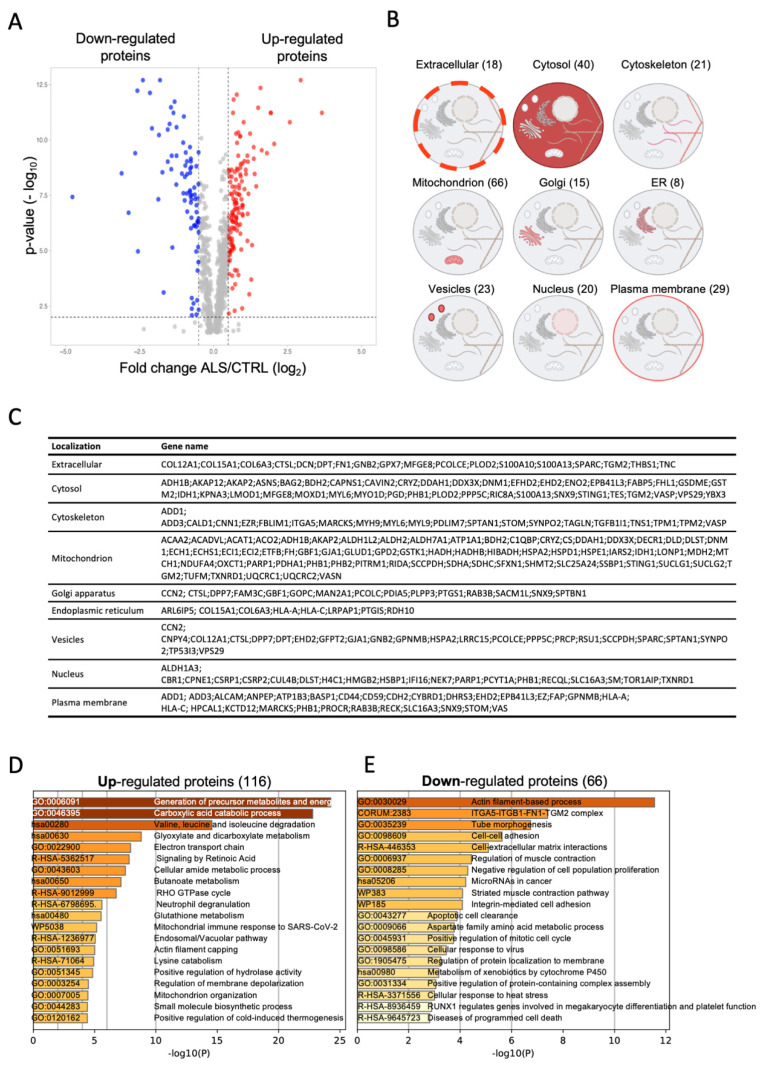
(**A**) Volcano plot of data from label-free quantification of proteins identified with at least two unique peptides (924 proteins) in amyotrophic lateral sclerosis (ALS) and control (CTRL) fibroblasts. Red and blue dots represent up- and down-regulated proteins with a log_2_ fold change ± 0.58 and *p*-value cut-off <0.05. (**B**) Extracellular and/or subcellular localization of differentially expressed proteins (DEPs). (**C**) List of differentially expressed proteins in ALS and CTRL fibroblasts. (**D,E**) Pathway enrichment analyses of up- and down-regulated DEPs, respectively. The 20 most significantly enriched pathways and biological processes in the proteomic data of DEPs were identified by Metascape. Enriched terms are colored by *p*-values. The x-axis shows the value of −log10(P). (**F**) Immunoblots showing the relative expression of TDP-43 and Lonp1 in CTRL and ALS fibroblasts data are reported as mean ± SEM of three independent blots. Values are the mean ± SEM of 3 independent experiments performed on fibroblasts from two CTRL subjects and the ALS proband * *p* < 0.05. (**G**) Immunoblots showing the intracellular distribution of TDP-43 and LONP1 in cytosolic (C), mitochondrial (M) and nuclear (N) fractions of CTRL and ALS fibroblasts. TOM-20 and LaminB1 show the purity of mitochondrial and nuclear fractions, respectively.

**Figure 3 ijms-23-11881-f003:**
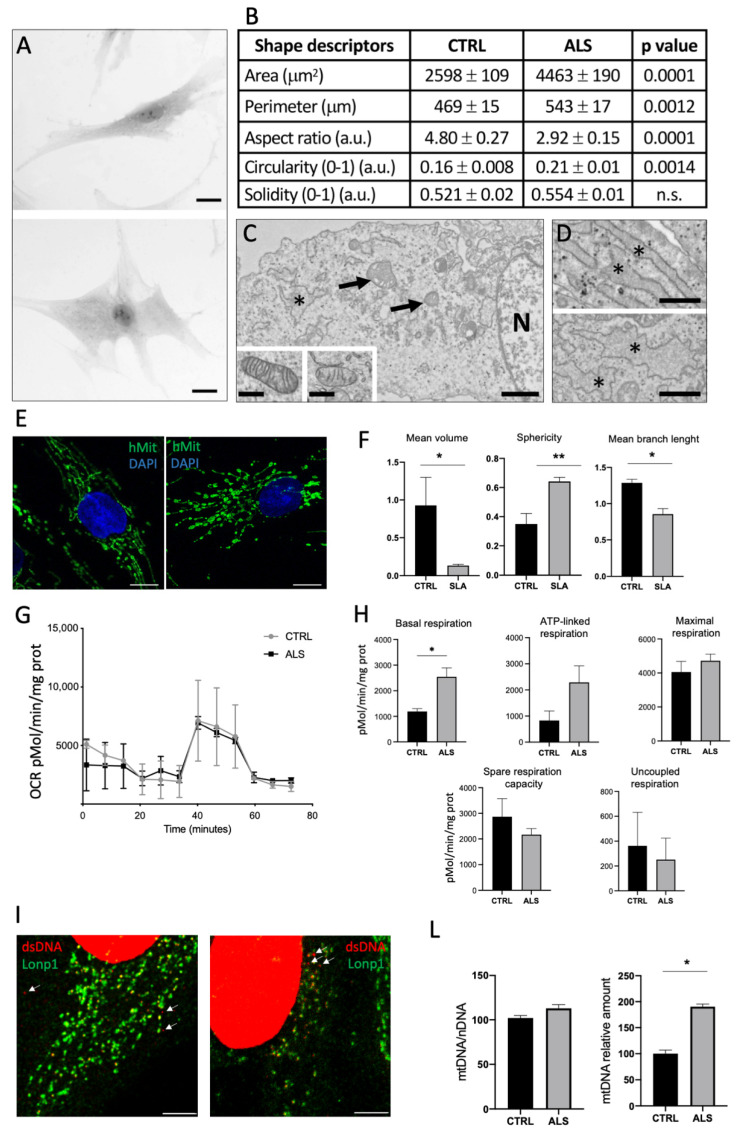
(**A**) Representative light microscopy images of fixed primary skin fibroblasts isolated from CTRL (up) and ALS (down) and stained with toluidine blue. Scale bar = 20 μm. (**B**) Quantification of area, perimeter, aspect ratio, circularity, and solidity are reported as mean ± SE. For each cell line 120 cells were analysed. a.u.= arbitrary unit. (**C**) Representative electron micrographs of ALS fibroblasts. N= nucleus; * = endoplasmic reticulum. Mitochondria are indicated by arrows. Scale bar = 1 μm. Inserts show a mitochondrion from CTRL (left) and from ALS fibroblast (right). Insert bar = 500 nm. (**D**) Representative electron micrographs of endoplasmic reticulum in CTRL (up) and in ALS (down) fibroblasts. Scale bar = 1 μm. (**E**) Representative confocal microscopy images showing mitochondrial topology in ALS and CTRL fibroblasts. Bar = 10 μm. (**F**) Mean volume of mitochondria, sphericity and mean branch length of mitochondria in CTRL and ALS fibroblasts. (**G**) Oxygen consumption rate of ALS and CTRL fibroblasts in basal conditions (min 0–20) and after injection of oligomycin (min 20–40), FCCP (min 40–60), and Antimycin A + Rotenone (min 60–80). Data have been normalized to protein content. (**H**) Basal respiration, ATP-linked respiration maximal respiration, spare respiration capacity and uncoupled respiration of fibroblasts from ALS and control. Data are expressed as pmol O_2_/min/mg prot and is a mean ± SD of two independent experiments. (**I**) Representative confocal microscopy image of a fibroblast from the proband and a healthy control labelled with dsDNA Ab and Lonp1 Ab. dsDNA outside mitochondria is indicated by arrows. Bar = 5 μm. (**L**) Levels of mtDNA normalized to nDNA, and relative amount of mtDNA in the cytosolic fraction in fibroblasts from the proband (ALS) and a healthy control (CTRL). Values are the mean ± SEM of 3 independent experiments performed on CTRL and ALS fibroblasts * = *p* < 0.05; ** = *p* < 0.01.

**Table 1 ijms-23-11881-t001:** Main patient’s symptoms and signs during disease course. * Medical Research Council’s (MRC) scale uses a score of 0 (no contraction) to 5 (normal power) to grade the power of a particular muscle group.

Timeline	Patient’s Symptoms	Neurological Examination
Eleven months after symptoms onset (at diagnosis)	Right hand muscles weakness Right hand muscles hypotrophy Widespread fasciculations and cramps Walking impairment	Severe weakness of intrinsic hand muscles (MRC * score 2 on the right and 3 on the left) Moderate weakness of foreharm muscles (MRC score 3–4). Intrinsic hand muscles atrophy Distal lower limbs weakness (MRC score 4–5) Widespread fasciculations Brisk tendon reflexes both at upper and lower limbs
Thirty-six months from symptoms onset	Dysarthria and dysphagia Pseudobulbar affect Upper and lower limbs weakness and atrophy Widespread fasciculations Orthopnoea requiring non-invasive ventilation a few months before	Dysarthria and tongue fasciculations. Mild weakness of soft palate Severe weakness of intrinsic hand muscles (MRC score 0 on the right and 1 on the left) Severe weakness of foreharm (MRC score 1–2) and arm (MRC score 2–3) muscles Bilateral foot drop and moderate proximal lower limbs weakness (MRC score 3) Distal limbs muscles atrophy Widespread fasciculation Brisk tendon reflexes at lower limbs
Forty months from symptoms onset	Anarthria, requiring eye-tracking communication system Dysphagia with nutrition possible only by percutaneous endoscopic gastrostomy Pseudobulbar affect Sialorrhea Severe tetraparesis with limbs atrophy Dyspnea and orthopnea requiring continuous invasive ventilation through tracheostomy	Anarthria; tongue weakness and fasciculations Palatal palsy Severe tetraparesis (MRC score 0–1) Widespread muscle atrophy Widespread fasciculations Brisk tendon reflexes at lower limbs with mild spasticity

## Data Availability

Proteomic data sets from CTRL and ALS fibroblasts will be available from the corresponding author upon request.

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
