# Peer review of "Mitochondrial and Endoplasmic Reticulum Alterations in a Case of Amyotrophic Lateral Sclerosis Caused by TDP-43 A382T Mutation"

_ijms, 2022, doi:10.3390/ijms231911881_

Round 1

Reviewer 1 Report

The case report by Giada Zanini et al. shows an ALS patient carrying a point mutation in the TARDBP gene, which encodes the protein TDP-43. The patient presents a family history of the mutant, with several relatives developing neurological symptoms. The authors isolated skin fibroblast from the patient and performed proteomics studies to detect the differentially expressed genes, founding a high fraction of differentially expressed genes in the mitochondria. The authors also validated some of the hits by immunoblot. The authors showed no differential expression of TDP-43 in the patient’s cells but differential sub-cellular localization in the mitochondria. Next, the authors evaluated different cellular parameters founding a different cellular morphology, mitochondria morphology, structure, and metabolism. Finally, the authors also detected mtDNA presence in the cytoplasm, correlated with an upregulated STING1. 

Overall, the work is very well conducted and characterized subcellular changes in the TDP-43 patient cells.

However, I have some major a minor comments and suggestions.

Major:

  • The authors mention in the title, the abstract, and the discussion of the ER alterations. However, there is no mention of this during the presentation of the results. This parameter is critical to clarify because otherwise, I suggest modifying the title of the manuscript. 
  • The text should discuss the possible mechanism, if this exists, about this point mutation leads to an increased mitochondrial localization. 
  • The study is compared to only one control cell, which also brings the concern of genetic background variability. I recommend including a second control and testing at least the main hits shown in Fig2E.

Minor

  • Please incorporate a table with all the described clinical symptoms.
  • The significant hits of proteomics should be in the main figure. 
  • In Fig2E, I recommend including a mitochondrial mass marker to confirm that the observed changes are because of changes in the protein levels and not in the mitochondrial mass.
  • In Fig3, the authors show cell morphology changes, attributing them to changes in the cytoskeleton protein. In this regard, it would be interesting to confirm these changes by staining the actin or the tubulin cytoskeleton. In this regard, this fibroblast may show an activated myofibroblast phenotype. Does the proteomics show any sign of this?
  • Please include control panels for TEM images. Also, include an inset of the images because the absence of cristae or cristae rearrangement from the panel is not clear.
  • In Panel C, the TEM images also show very dilated ER. Is this a common feature in the patient’s cells? In this regard, quantification of the ER-mitochondria distance and cristae structure would provide more clues to the field about how a TDP-43 mutant affects the mitochondria.
  • It is not clear from the confocal images of the dsDNA dots. Please use a combination of colors with better contrast. 
  • Please indicate the molecular weight in the immunoblots.

Author Response

Dear Editor,

I have the pleasure to send you the revised version of the manuscript entitled “Mitochondrial and Endoplasmic reticulum alterations in a case of Amyotrophic Lateral Sclerosis caused by TDP-43 A382T mutation”  by G. Zanini et al., to be considered for publication on the International Journal of Molecular Sciences. The manuscript has been carefully amended, based on the comments and suggestions of the reviewers. Here below, the point-to-point answers to the comments of the Rev #1 are reported:

REVIEWER #1

The case report by Giada Zanini et al. shows an ALS patient carrying a point mutation in the TARDBP gene, which encodes the protein TDP-43. The patient presents a family history of the mutant, with several relatives developing neurological symptoms. The authors isolated skin fibroblast from the patient and performed proteomics studies to detect the differentially expressed genes, founding a high fraction of differentially expressed genes in the mitochondria. The authors also validated some of the hits by immunoblot. The authors showed no differential expression of TDP-43 in the patient’s cells but differential sub-cellular localization in the mitochondria. Next, the authors evaluated different cellular parameters founding a different cellular morphology, mitochondria morphology, structure, and metabolism. Finally, the authors also detected mtDNA presence in the cytoplasm, correlated with an upregulated STING1. 

Overall, the work is very well conducted and characterized subcellular changes in the TDP-43 patient cells.

However, I have some major a minor comments and suggestions.

ANSWER: We thank the reviewer for her/his comments, and for appreciating our report.

Major:

The authors mention in the title, the abstract, and the discussion of the ER alterations. However, there is no mention of this during the presentation of the results. This parameter is critical to clarify because otherwise, I suggest modifying the title of the manuscript. 

ANSWER: We thank the reviewer for this comment. As requested, we added a few lines in the Results section where results concerning ER are reported:

“Figures 3C and 3D show the main ultrastructural abnormalities observed in ALS fibro-blasts. Mitochondria showed an elevated heterogeneity in morphology: some were particularly enlarged with distorted or excessively branched cristae, others were char-acterized by reduction of many cristae (Figure 3 C, insert). Moreover, in comparison with CTRL fibroblasts (Figure 3D), the endoplasmic reticulum cisterns in ALS cells were swollen with an irregular shape, suggesting an accumulation of unfolded/misfolded polypeptides in the lumen (Figure 3E).”

Furthermore, we discussed this point a little bit more in detail in the discussion section:

“The alteration in the shape of A382T fibroblasts, never described before, likely reflects the changes in the expression of cytoskeleton-related pathways, and the impairment in the ER and mitochondrial ultrastructure we observed. Although only a few proteins in the ER appeared differentially expressed in ALS fibroblasts, the ER appeared enlarged, irregularly shaped and swollen, likely because of the accumulation of misfolded proteins,  and is suggestive of ER stress. ER stress has been often observed in ALS, and the presence of  abnormal protein aggregates and/or of inclusions of misfolded proteins is a commone feature of ALS; the presence of an enlarged ER could be of particular interest because, differently from other pathogenetic mutations like M337V, A382T causes the formation of cytoplasmic TDP-43 aggregates only in the presence of ER stress [20]. As the presence of enlarge ER was not reported before in A382T fibroblasts, these observations further underlined the heterogeneity in the phenotypical manifestations of p.Ala382Thr mutation.“

  • The text should discuss the possible mechanism, if this exists, about this point mutation leads to an increased mitochondrial localization. 

ANSWER:   A pivotal paper by Wang et al. (Nat Med 2016) showed that mitochondrial import of TDP-43 depends on the presence of internal mitochondrial localization motifs. The authors identified six putative internal motifs (M1-M6) and demonstrated that two of them (M1 and M3) did favour mitochondrial import of TDP-43, being M1 the most efficient. Neither M1 nor M3 include the codon 382. However, it must be noted that A382T mutations falls in the M6 sequence.  Although there is no direct proof that M6 motif is actually functional, we can speculate that Ala>Thr mutation in M6 could make it more efficient. An alternative hypothesis could be that this mutation indirectly modifies the accessibility of M1 or M3 sequence during TDP-43 import.  These considerations have been added to the discussion:

“We first confirmed that p.Ala382Thr mutation determines a higher level of TDP-43 accumulation in the mitochondria. It has been previously shown that mitochondrial import of TDP43 depends on the presence of internal mitochondrial localization motifs [27,28]. Wang et al. identified six putative internal mitochondrial import motifs (M1-M6) in TDP-43 and demonstrated that at least two of them (M1 and M3) are functional, being M1 the most efficient. Neither M1 nor M3 include the codon 382 [28]. However, the p.Ala382Thr mutations falls in the M6 motif. Although there is no direct proof that M6 is functional, we can speculate that p.A382T mutation in M6 makes it more efficient. An alternative, non-mutually exclusive hypothesis could be that this mutation indirectly modifies the accessibility of M1 or M3 sequence during TDP-43 import.”

and two references have been added:

  1. Francois-Moutal, L.; Perez-Miller, S.; Scott, D.D.; Miranda, V.G.; Mollasalehi, N.; Khanna, M. Structural Insights Into TDP-43 and Effects of Post-translational Modifications. Front Mol Neurosci 2019, 12, 301, doi:10.3389/fnmol.2019.00301.

  1. Wang, W.; Wang, L.; Lu, J.; Siedlak, S.L.; Fujioka, H.; Liang, J.; Jiang, S.; Ma, X.; Jiang, Z.; da Rocha, E.L.; et al. The inhibition of TDP-43 mitochondrial localization blocks its neuronal toxicity. Nat Med 2016, 22, 869-878, doi:10.1038/nm.4130.

  • The study is compared to only one control cell, which also brings the concern of genetic background variability. I recommend including a second control and testing at least the main hits shown in Fig2E.

ANSWER: We added a second control for figure 2E (now Figure 2F), starting from protein extracts previously obtained from fibroblasts of a healthy donor. Because of time and ethical constrains in collecting and using samples, we limited the analysis just to the proteins analyzed in the Figure 2E. The new WB is reported in the supplementary figures, while WB graphs have been updated with this new data, which do not modify the conclusions drawn.

Minor

  • Please incorporate a table with all the described clinical symptoms.

ANSWER: As requested, we included a table (Table 1) with the described clinical symptoms.

  • The significant hits of proteomics should be in the main figure. 

ANSWER: we moved the significant hits (Supplementary Table 3) in the main figure (Figure 2, panel C), as requested.  

  • In Fig2E, I recommend including a mitochondrial mass marker to confirm that the observed changes are because of changes in the protein levels and not in the mitochondrial mass.

ANSWER: we included a WB of anti hMit antibody (Sigma Aldrich) showing no significant change.   The WB is now included in the Figure 2F.

  • In Fig3, the authors show cell morphology changes, attributing them to changes in the cytoskeleton protein. In this regard, it would be interesting to confirm these changes by staining the actin or the tubulin cytoskeleton. In this regard, this fibroblast may show an activated myofibroblast phenotype. Does the proteomics show any sign of this?

ANSWER:  We thank the reviewer for this precious suggestion.  We inspected the proteomics dataset, looking for specific markers of myofibroblasts, but we did not find significant hits that could suggest an activated myofibroblast phenotype. In particular, we did not observe significant variation of ASMA –alpha smooth muscle actin, typically expressed in myofibroblasts – and other proteins associated with myofibroblast phenotype, such as COL11A1, LOXL2, CTGF and the transcription factor GLI2. Thus, in our opinion, there is no clear evidence of myofibroblast activation.

Please include control panels for TEM images. Also, include an inset of the images because the absence of cristae or cristae rearrangement from the panel is not clear.

ANSWER:  As requested, we added we added an inset of a mitochondrion from ALS patient showing cristae alterations in a representative mitochondrion from ALS, in comparison to a CTRL mitochondrion. Accordingly, these lines have been added to the manuscript:

“Figures 3C and 3E show the main ultrastructural abnormalities observed in ALS fibroblasts. Mitochondria showed an elevated heterogeneity in morphology: some were particularly enlarged with distorted or excessively branched cristae, others were characterized by reduction of many cristae (Figure 3 C, insert). Moreover, in comparison with CTRL fibroblast (Figure 3D) the endoplasmic reticulum cisterns in SLA cells were swollen with an irregular shape suggesting an accumulation of unfolded/misfolded polypeptides in the lumen (Figure 3E).”

  • In Panel C, the TEM images also show very dilated ER. Is this a common feature in the patient’s cells? In this regard, quantification of the ER-mitochondria distance and cristae structure would provide more clues to the field about how a TDP-43 mutant affects the mitochondria.

ANSWER: The reviewer is right: the dilated ER is a common feature of the patient’s cells. We added an inset (Figure 3E) showing another region with dilated ER, in comparison to normal ER from CTRL cells.   Concerning the analysis of ER-mitochondria distance and cristae structure, this could be and interesting data, it was not possible to be performed in a reliable form on the images we obtained with TEM. 3D electron tomography technique with serial section should be the technical approach of choice to get reliable data, and we unfortunately did not have access to this technology.

  • It is not clear from the confocal images of the dsDNA dots. Please use a combination of colors with better contrast. 

ANSWER: We have increased contrast (without altering the image) to make dsDNA dots more evident.  

Please indicate the molecular weight in the immunoblots.

ANSWER: as requested, we added the MW in the blots, showing the MW ruler.

We thank again the reviewer for his/her comments, which helped us to improve the quality and clarity of our report, and we hope that now the manuscript can be considered suitable for publication in the International Journal of Molecular Sciences. 

Kind regards, 

Dr. Marcello Pinti

Reviewer 2 Report

The case report by Zanini et al. entitled by “Mitochondrial and Endoplasmic reticulum alterations in a case of Amyotrophic Lateral Sclerosis caused by TDP-43 A382T mutation” and aims at evaluating the effect of TDP-43 A382T mutation on amyotrophic lateral sclerosis, the study studied the familiar/genetic impact, mainly through mitochondrial dysfunctions. The manuscript is interesting, and in the scopus of the journal International Journal of Molecular Sciences. Overall, the manuscript is well written, presented and discussed. However, I recommend some minor revision noted in the following.

The English language should be revised as often grammatical errors are encountered.

Avoid the use of abbreviations in the abstract and in keywords. It is desirable to define the terms in other parts of the manuscript.

The authors should be improved the introduction, for example consequences of mutation (lines 53-57).

Author Response

Dear Editor,

I have the pleasure to send you the revised version of the manuscript entitled “Mitochondrial and Endoplasmic reticulum alterations in a case of Amyotrophic Lateral Sclerosis caused by TDP-43 A382T mutation”  by G. Zanini et al., to be considered for publication on the International Journal of Molecular Sciences. The manuscript has been carefully amended, based on the comments and suggestions of the reviewers. Here below,  the point-to-point answers to the comments of the Rev #2 are reported:

The case report by Zanini et al. entitled by “Mitochondrial and Endoplasmic reticulum alterations in a case of Amyotrophic Lateral Sclerosis caused by TDP-43 A382T mutation” and aims at evaluating the effect of TDP-43 A382T mutation on amyotrophic lateral sclerosis, the study studied the familiar/genetic impact, mainly through mitochondrial dysfunctions. The manuscript is interesting, and in the scopus of the journal International Journal of Molecular Sciences. Overall, the manuscript is well written, presented and discussed. However, I recommend some minor revision noted in the following.

ANSWER: We thank the reviewer for her/his comments, and for appreciating our report. 

i) The English language should be revised as often grammatical errors are encountered.

ANSWER: the manuscript has been revised by an English native speaker 

ii) Avoid the use of abbreviations in the abstract and in keywords. It is desirable to define the terms in other parts of the manuscript.

ANSWER: we removed abbreviations from the abstract, as requested.

iii) The authors should be improved the introduction, for example consequences of mutation (lines 53-57).

ANSWER:  As requested, we expanded the introduction, which is now come comprehensive, and includes a short paragraph focused on the consequences of mutations:

“The majority of the mutations affect the C-terminal glycin-rich domain of the protein [3,4] (…) The effects of TDP-43 pathogenetic mutations on TDP-43 functions are quite heterogenous. A large fraction of the pathogenetic TDP-43 mutations, including A382T, has been shown to decrease the nuclear localization of the protein, to increase its phosphorylation status, and to favour the formation of C-terminal fragments deriving from proteolytic digestion of the full-length proteins [11]. These fragments can form insoluble aggregates; however, the A382T mutation is not associated with an increased insolubility of the full-length protein, nor reduces the solubility of the C-terminal fragments. While some pathogenic mutations, such as M337V, spontaneously mislocalizes to the cytoplasm and form insoluble aggregates, A382T mutation leads to aggregate formation only in the presence of ER stress. A minority of mutations, which does include A382T, also increases the protein half-life [11].

We also added new references to support these pieces of information:

  1. Pesiridis, G.S.; Lee, V.M.; Trojanowski, J.Q. Mutations in TDP-43 link glycine-rich domain functions to amyotrophic lateral sclerosis. Hum Mol Genet 2009, 18, R156-162, doi:10.1093/hmg/ddp303.
  2. Wood, A.; Gurfinkel, Y.; Polain, N.; Lamont, W.; Lyn Rea, S. Molecular Mechanisms Underlying TDP-43 Pathology in Cellular and Animal Models of ALS and FTLD. Int J Mol Sci 2021, 22, doi:10.3390/ijms22094705.
  3. Watanabe, S.; Kaneko, K.; Yamanaka, K. Accelerated disease onset with stabilized familial amyotrophic lateral sclerosis (ALS)-linked mutant TDP-43 proteins. J Biol Chem 2013, 288, 3641-3654, doi:10.1074/jbc.M112.433615.

We thank again the reviewer for her/his comments, and we hope that now the manuscript can be considered suitable for publication on the  International Journal of Molecular Sciences. 

Kind regards, 

Dr. Marcello Pinti